# Pixelated Filter Array for On-Chip Polarized Spectral Detection

**DOI:** 10.3390/nano13192624

**Published:** 2023-09-23

**Authors:** Yuechen Liu, Chao Feng, Siyu Dong, Jingyuan Zhu, Zhanshan Wang, Xinbin Cheng

**Affiliations:** 1MOE Key Laboratory of Advanced Micro-Structured Materials, Shanghai 200092, China; 2130941@tongji.edu.cn (Y.L.);; 2Institute of Precision Optical Engineering, School of Physics Science and Engineering, Tongji University, Shanghai 200092, China; 3Shanghai Frontiers Science Center of Digital Optics, Shanghai 200092, China; 4Shanghai Institute of Intelligent Science and Technology, Tongji University, Shanghai 200092, China

**Keywords:** polarized spectral detection, filter array, on-chip detection, metasurface

## Abstract

On-chip multi-dimensional detection systems integrating pixelated polarization and spectral filter arrays are the latest trend in optical detection instruments, showing broad application potential for diagnostic medical imaging and remote sensing. However, thin-film or microstructure-based filter arrays typically have a trade-off between the detection dimension, optical efficiency, and spectral resolution. Here, we demonstrate novel on-chip integrated polarization spectral detection filter arrays consisting of metasurfaces and multilayer films. The metasurfaces with two nanopillars in one supercell are designed to modulate the Jones matrix for polarization selection. The angle of diffraction of the metasurfaces and the optical Fabry–Perot (FP) cavities determine the spectrum’s center wavelength. The polarization spectral filter arrays are placed on top of the CMOS sensor; each array corresponds to one pixel, resulting in high spectral resolution and optical efficiency in the selected polarization state. To verify the methodology, we designed nine-channel polarized spectral filter arrays in a wavelength range of 1350 nm to 1550 nm for transverse electric (TE) linear polarization. The array has a 10 nm balanced spectral resolution and average peak transmission efficiency of over 75%, which is maintained by utilizing lossless dielectric material. The proposed array can be fabricated using overlay e-beam lithography, and the process is CMOS-compatible. The proposed array enables broader applications of in situ on-chip polarization spectral detection with high efficiency and spectral resolution, as well as in vivo imaging systems.

## 1. Introduction

Multi-dimensional optical field detection has attracted significant attention because it allows one to capture highly detailed information. Polarization spectral detection, which enables the simultaneous measurement of spatial, polarized, and spectral information, substantially enhances the dimensions of the data collected. Conventional polarization spectral imagers are based on discrete optical components, resulting in bulky, heavy, and complex optical systems with low stability. On-chip integration is a prevailing trend in optical device development. Miniaturized and integrated devices can be integrated into intelligent electronic platforms like unmanned aerial vehicles, handheld devices, and mini-satellites, enabling field inspections or real-time monitoring. Such devices can be applied in various fields, such as diagnostic medical imaging [1,2], remote sensing [3], target detection [4], aerosol detection [5], agro-environmental sciences [6,7], and so on. Compared to spectral detection alone, polarized spectral detection, which measures polarization information, can enhance a device’s identification capabilities in complex environments. Moreover, this detection method relies on identifying specific chemical components through spectral feature peak recognition. To ensure detection accuracy, it places high demands on spectral resolution.

In recent years, research based on various type of microstructures has enabled a wide range of optical modulation functions [8,9,10,11,12,13]. Various filter arrays based on micro/nanostructures have been integrated on top of image sensors, with a pitch matching the pixel size to improve the integration and miniaturization of on-chip devices [14,15,16,17,18,19,20]. The anisotropic meta-atoms in phase-modulation metasurfaces are spatially multiplexed to obtain multiple polarization channels with different phase profiles [21,22,23,24,25]. Incident light with different polarization states is reflected into different space domains while the spectral information is obtained simultaneously due to the off-axis dispersion. These designs enable on-chip polarization spectral detection. However, a certain optical path is required for high spectral resolution, limiting improvements to further integration. Resonant microstructures are more compatible with on-chip integration [26,27,28,29,30,31,32]. The transmission of light with different center wavelengths and polarization states can be regulated by modulating the period, duty cycle, and anisotropy. However, these approaches typically have low performance. The high absorption loss and resonance bandwidth of metal structures result in low transmittance and spectral resolution. In contrast, dielectric resonance structures have higher transmittance performance but cannot distinguish different polarization states. On-chip polarized spectral detection has high requirements for spectral transmittance and cutoff performance to improve the detection signal/noise ratio and spectral resolution in practical applications.

The Fabry–Perot (FP)-type cavity is a classic scheme for on-chip spectral imaging [33,34,35,36,37,38]. Due to the narrow pass-band, wide cutoff band, and customizable transmittance peak, a relatively higher spectral resolution can be obtained using the FP cavity compared with a filter based on micro/nanostructures. By modifying the optical thickness of the spacer layer, different passbands can be obtained for spectroscopic measurement. However, due to the polarization insensitivity of isotropic optical films, the FP cavity cannot be used for polarized spectral detection. Therefore, it is possible to realize the polarization and spectral measurements simultaneously by combining FP cavity with micro-nano structures with the capability of polarization modulation. The notion of adding filter arrays based on micro/nanostructures to the spacer layer of the FP-type cavity has been investigated. The inclusion of high-index nanopillar arrays adds a phase shift to the cavity and leads to a subsequent shift in the resonant peak [19,39]. Then, by adding anisotropy to the nanostructures, the FP cavity becomes sensitive to polarization [40]. However, reconstruction via the use of algorithms must be applied to recover the polarization spectrum, causing low detection speed and insufficient measurement accuracy.

In this work, we propose novel on-chip integrated polarization spectral detection filter arrays consisting of metasurfaces and multilayer films. The metasurfaces and multilayer films are cooperatively modulated through angular coupling between them. The metasurface exhibits high polarization and phase manipulation performance, and the multilayer film FP cavity enables amplitude regulation with high spectral performance. The target polarization is selected by modulating the Jones matrix utilizing the interference between dual meta-atoms. Distinct wavelength channels are created by varying the incident angle instead of modulating the FP cavity itself by designing the phase gradient of the metasurface. As a proof of concept, we design a nine-channel polarization spectral filter device that enables simultaneous spectral detection and transverse electric (TE) linear polarization detection with a favorable transmission ratio of over 75% and ~10 nm spectral resolution. Finally, we discuss the on-chip integrated alignment method and standard semiconductor fabrication process of the proposed multi-dimension detection filter via overlay e-beam lithography, physical vapor deposition (PVD), and reactive ion etching (RIE) to improve its potential for practical application.

## 2. Theory and Simulation Result

Figure 1 shows a schematic of the proposed polarization spectral filter array. The top layer consists of mosaic metasurfaces which act as a polarization filter and diffraction element. When the metasurfaces are illuminated by two orthogonal linear polarizations, they maintain a high transmission rate for the desired polarization state while reflecting the other state. After passing through the metasurface, the polarized light is diffracted to a specific angle and enters the multilayer film below. The multilayer film is composed of a spacer layer on top to prevent guided-mode resonance and the FP cavity below as a wavelength filter, which filters the diffracted light incident from the metasurface layer so that only narrow-band polarized light can pass through. Different filtered polarization states and diffraction angles of the metasurface are designed for each pixel in the array, resulting in multiple polarization and wavelength channels. The effective thickness of the FP cavity can be changed by adjusting the incident angle of the FP cavity, which is related to the diffraction angle, so that a specific wavelength can be selected utilizing a single FP cavity.

### 2.1. Design of Polarization Selective Metasurface

The optical response of the metasurface can be expressed by using a Jones matrix, namely, $J=\left[\begin{array}{cc} A_{\mathrm{xx}}e^{{i\varphi }_{\mathrm{xx}}} & A_{\mathrm{xy}}e^{{i\varphi }_{\mathrm{xy}}}\\ A_{\mathrm{yx}}e^{{i\varphi }_{\mathrm{yx}}} & A_{\mathrm{yy}}e^{{i\varphi }_{\mathrm{yy}}} \end{array}\right]$, where x and y represent a pair of orthogonal linear polarization states, and A_ij_ and φ_ij_ (i, j = x, y) are the amplitude and phase responses from the i-polarized input light to the j-polarized output light. We consider a dielectric metasurface composed of individual anisotropic nanopillars [41,42] with a rotation angle θ = 0. The Jones matrix can be simplified to $J=\left[\begin{array}{cc} A_{x}e^{{i\varphi }_{x}} & 0\\ 0 & A_{y}e^{{i\varphi }_{y}} \end{array}\right]$, where A_i_ and φ_i_ (i = x, y) represent the amplitude shift and transmission phase change of the incident x- or y-polarized light. If the metasurface only allows x-polarized light to pass through, ideally, the matrix should satisfy $\left|A_{x}e^{{i\varphi }_{x}}\right|=1$ and $\left|A_{y}e^{{i\varphi }_{y}}\right|=0$. However, it is difficult to develop a universal approach to design such a metasurface due to the high transmission rate of dielectric material, the conventional modulating methods for dielectric materials based on Mie resonance, etc., limiting the structural degrees of freedom and making it difficult to additionally modulate the polarization on this basis.

A supercell consisting of multiple anisotropic nanopillars is used to overcome the modulation limit of the Jones matrix of the meta-atom metasurface [43,44,45]. If no coupling exists between the nanopillars, the total transmission rate of each nanopillar can be expressed as $E_{\mathrm{out}}=\underset{i=1}{\sum }E_{\mathrm{outi}}\;,$ where i is the index of the i-th nanopillar. Therefore, the Jones matrix of the supercell can be defined as
(1)$$J=\underset{i=1}{\sum }{\;J}_{i}=\underset{i=1}{\sum }\;\left[\begin{array}{cc} A_{x_{i}}e^{{i\varphi }_{x_{i}}} & 0\\ 0 & A_{y_{i}}e^{{i\varphi }_{y_{i}}} \end{array}\right]$$
where J_i_ is the Jones matrix of the i-th nanopillar subunit. The polarization filtering metasurface must satisfy $|\sum A_{x}e^{{i\varphi }_{x}}|=1$ and $|\sum A_{y}e^{{i\varphi }_{y}}|=0$. For the multi-nanopillar supercell, the amplitude and phase terms of each polarization component of the Jones matrix can be obtained by solving a multivariate system. Mathematically, at least four independent variables are needed to obtain all four dependent variables. Each anisotropic nanopillar has up to three variables (i.e., φ_x_, φ_y_, θ) [42]. At least two individual nanopillars are needed to achieve arbitrary modulation. We consider a supercell consisting of two subunits. This configuration contains two nanopillars that exhibit a fixed transmission coefficient of 1. The polarization-dependent phase response is as follows. The phase response remains identical when the surface is illuminated with x-polarized light and varies by π when illuminated with y-polarized light. Consequently, the conditions $|\sum A_{x}e^{{i\varphi }_{x}}|=1$ and $|\sum A_{y}e^{{i\varphi }_{y}}|=0$ are satisfied, and the polarization filter can be designed. Figure 2a shows a schematic of this polarization unit.

In our design strategy, the propagation phase changes φ_x_ and φ_y_ are achieved by varying the duty cycle L_x_ and L_y_. The design is straightforward. We create a database of the propagation phases (φ_x_ and φ_y_) and transmission efficiencies (T_x_ and T_y_) by scanning the dimensions, and we streamline and reorganize the database into the form of L_x_ (φ_x_, φ_y_), L_y_ (φ_x_, φ_y_), (Figure 2b). This strategy facilitates data retrieval. We obtain the structure according to the target transmission rate and phase response. Figure 2d shows an example of the polarization unit and its polarized spectrum at wavelengths 1.35–1.55 μm. We utilize silicon attributed to the high index and acceptable absorptivity(n_Si,1450nm_ = 3.381, k_Si,1450nm_ = 1.83 × 10^−3i^). The unit cell has lattice constant L = 650 nm, which is about half the wavelength, avoiding higher order diffraction. Each nanopillar has a height of h = 1 μm, which ensures sufficient modulation capability without introducing too many difficulties in fabrication. The structure shows excellent polarization selective performance.

Next, we combine the polarization selection with the diffraction of the metasurface to modulate the incident angle of the FP cavity for multi-wavelength selection. According to Huygens’ principle, when an electromagnetic wave is vertically incident on a metasurface, the wavefront can be obtained by changing the phase of each meta-atom as a subwavelength source. By varying the lateral dimensions and effective indices, a 2π-period phase gradient can be obtained to achieve high diffraction efficiency. The selection of the polarization state and diffraction and the phase profile is shown in Figure 2e. The phase response of the meta-atoms to the transmitted polarization state results in a phase gradient with a 2π period. It must be individually tailored to different target wavelength channels. The phase profile of the filtered polarized light forms multiple interleaved supercells composed of a pair of π-phase-varied subunits. The broadband efficiency is high, and the distribution of the electromagnetic fields indicates high polarization selection performance.

### 2.2. Design of Coupled Metasurface and FP Cavity

The FP cavity is a compact color filter consisting of two mirrors with a spacer layer between. Incident light is emitted between the spacer layers after repeated interference between the upper and lower mirrors, and high resolution can be obtained if the reflectivity of the mirrors is sufficiently high. In dielectric FP multilayers, the mirrors on both sides consist of a stack of highly reflective films. The transmission band and its central wavelength in the FP cavity can be described by the following equation
(2)$$\lambda =\frac{{2\mathrm{nLcos}\beta }_{0}}{m+\left(\varphi_{1}{+\varphi }_{2}\right)∕2\pi }m=0,\;\pm 1\ldots$$
where n and L are the refractive index and physical thickness of the spacer layer, respectively, $\beta_{0}$ is the incident angle, m is the order of transmission, and φ is the reflection phase (RP) of the two mirrors, which is approximately equal. The transmission center wavelength varies with the incident angle; thus, the selection of multiple wavelengths for a single FP multilayer film can be achieved by tuning the incident angle, as shown in Figure 3a.

The FP multilayer film is designed utilizing hydrogen silicon (SiH, H) and SiO_2_ (L); both have low absorptions in the near-infrared band. The thin films consist of (LH)_3_L(HL)_3_. Three sets of highly reflective film stacks are placed on both sides of the spacer layer to ensure high spectral resolution. The lower refractive index of SiO_2_ in the spacer layer enables a wider cutoff band to ensure a higher spectral bandwidth range. It should be noted that the cutoff band shifts with a change in the incident angle. Thus, it is necessary to prevent the cutoff band from overlapping with other spectral transmission peaks when designing the FP cavity.

The angular coupling between the metasurface and the FP multilayer film must be modulated to achieve the selection of light emitted at the targeted wavelength. When light passes through the metasurface, the dispersion of the structure results in an inability to maintain a stable 2π phase, resulting in various angles that depend on the phase gradient of the metasurface. The blue curve in Figure 3b shows the relationship between the wavelength and the diffraction angle, where $\sin\theta_{t}-\sin\theta_{i}=\frac{\lambda_{0}}{2\pi }\frac{d\Phi }{\mathrm{dx}}$, θ_t_ and θ_i_ are the transmittance and incident angles, respectively, λ_0_ is the wavelength of incident light, and $\frac{d\Phi }{\mathrm{dx}}$ is the phase gradient of the metasurface. Next, the light propagates through the FP structure, and only specific wavelengths can be transmitted depending on the incident angle. This relationship is expressed by Equation (2), shown by the green curve in Figure 3b. Only one intersection point occurs, indicating that only one wavelength of the light diffracted by the metasurface matches the transmission condition of the FP cavity. We can filter specific wavelengths by modulating the wavelength at this intersection. First, calculate the FP incident angle corresponding to this target wavelength. Based on this angle, the phase gradient can be computed to calculate suitable periods for the metasurface and subunits. Then, scan the metasurface transmission phase based on the determined geometric parameter at the target wavelength. We can obtain the solution for the structure combined with phase distributions for both polarizations. Scanning the database at their respective target wavelengths in different channels ensures that the metasurface is not affected by dispersion, thereby maintaining the highest diffraction efficiency (Figure 2e(iii)) and achieving the device’s optimal transmission efficiency.

Based on the theoretical design and particle swarm optimization, the spectral performance of the on-chip polarized spectral detector with a spacer layer was evaluated. The left and right figures represent the transmittance of the TE-polarized light and the TM-polarized light of multiple channels, respectively. In the left image (Figure 4a), the TE-polarized light detection with multiple channels was achieved in the bandwidth range of 1.35–1.55 μm. The nine different colors represent the transmission of nine wavelength channels. Due to the utilization of an all-dielectric structure, the peak transmittance can reach above 0.9, with an average exceeding 0.75. However, there are still some efficiency losses. These losses are due to the periodicity of the upper-layer metasurface that exceeds subwavelength scales. This leads to the presence of higher-order diffraction and results in a reduction in diffraction efficiency. By observing the transmission curves at 1.525 μm and 1.55 μm, we can find shorter peaks on the left side of the main peak that correspond to higher-order diffraction. Furthermore, we can also observe slight fluctuations in the curves, which are caused by the coupling between the sub units of the metasurface. The spectral resolution (full width at half maximum, FWHM) of the device is less than 10 nm (determined by using the Bragg reflectors on both sides of the FP cavity); the higher the reflectance, the narrower the FWHM. The right image (Figure 4b) shows the transmittance of the device under TM polarization. The colors of different channels correspond one-to-one with those in the left image. It can be observed that the transmission of TM-polarized light is suppressed but still exhibits slight residuals. This is attributed to coupling between subunits and errors generated during transmission phase simulation and data fitting processes. A SiO_2_ layer with a thickness of 2 um is placed between the metasurface and the FP cavity to prevent the guided-mode resonance from reducing the spectral performance.

Similarly, the detection of other orthogonal linear polarization states can be achieved by rotating each metasurface to the angle parallel to the polarization direction. Based on this design, an imaging array can be created for polarized spectral detection using multiple polarization and wavelength channels in a mosaic pattern. 

## 3. Discussion

We have presented a novel design of an on-chip polarized spectral detector by combining a multilayer film with a microstructure. The proposed quasi-3D polarized spectral detector has more design freedom and better polarization amplitude modulation capability due to the combination of the FP cavity and the microstructure, resulting in high efficiency and spectral resolution for on-chip detection. The spectral resolution of the device depends on the FWHM of the FP transmission peaks. The FWHM is related to the reflectivity of the Bragg mirrors, which can be changed by increasing or decreasing the layers of Bragg mirrors on each side of the spacer layer. The spectral resolution (FWHM) can be described as follows:(3)$$\frac{2∆\lambda }{\lambda_{c}}=\frac{2}{m+(\varphi_{1}+\varphi_{2})/2\pi }\sin^{-1}\left(\frac{1-R}{2\sqrt{R}}\right)$$
where 2∆λ is the FWHM, R is the reflectivity of the mirrors, and $\lambda_{c}$ is the central wavelength. The FWHM can be narrowed to obtain a greater spectral resolution for the detection of finer spectral information or widened to improve energy utilization for higher optical efficiency. The polarization spectrum is not the only solution since we can obtain different operating bands by changing the material and geometric parameters of the structure.

Although our design achieved high efficiency, efficiency loss and inter-pixel crosstalk may occur due to the angle of light emission. The light gradually deviates from the position of the detector pixels aligned directly below during the propagation. As this angle increases, the light may be projected onto adjacent pixels. If the size of the pixels is relatively small and the distance between the device and the detector is relatively large, a large proportion of the total light intensity will not incident on the correct pixel, causing substantial errors in spectral transmittance. These errors can be reduced by optimizing the on-chip integration process to reduce the spacing between the device and the detector image source. In addition, the errors can be corrected by further integrating metasurfaces for calibration. We can add a layer of metasurface arrays at the bottom of the multilayer film, which can be metalens or phase gradient anomalous refraction grating, with each metasurface in the array matching a pixel in the upper metasurface. By reshaping the wavefront, we can redirect light back to the original position to correct angular errors. The approach required to correct those errors will be studied in the future.

## 4. Fabrication Method

To widen the practical applications of the proposed polarized spectral detector, we describe the on-chip integrated alignment method and CMOS-compatible fabrication process via e-beam lithography, physical vapor deposition (PVD), and reactive ion etching (RIE), as shown in Figure 5.

The multilayer polarized spectral filter arrays were fabricated on the pixel area. The electrodes on the detector must be protected during fabrication, and each image pixel must be exposed. Overlay e-beam lithography was carried out using gold alignment markers at the four corners of the detector to improve the positional accuracy of the protective layer without obscuring the image pixels. The photoresist PMMA was spin-coated using a large beam current and beam step size to save time. To prevent the detector from being damaged by high temperatures, we employed vacuum baking during the spin-coating process to reduce the volatility temperature of the photoresist solvent. The sample was developed in a methyl isobutyl ketone (MIBK):IPA (1:3) solution and rinsed with IPA, forming a protective layer on the electrodes.

A multilayer FP film and a Si layer were deposited on the surface of the detector via PVD. The etch-resistant and high-resolution photoresist ZEP 520A was chosen as the soft etching mask. After spin-coating and heating on a hotplate, e-beam exposure of the meta-atoms structures was carried out via overlay e-beam lithography. The sample was developed in pentyl-acetate and rinsed with IPA. The patterned photoresist acted as a soft mask during the subsequent RIE to etch the opened Si on the sample. A mixture of CHF_3_ and O_2_ was applied during RIE. During this process the primary role of CHF_3_ is to provide fluorine atoms to etch the silicon, and its secondary role is to provide carbon to generate a passivation layer on the sidewalls to ensure the steepness of the etched structure. The role of oxygen is to generate an oxide layer that increases the anisotropy and can increase the etching rate. Then, the remaining resist was removed by O_2_ plasma. Finally, the photoresist was dissolved in acetone, and the protective layer on the electrodes was removed, leaving the multilayer structures and obtaining the on-chip polarized spectral imager.

## 5. Conclusions

We proposed new on-chip integrated polarization spectral detection filter arrays consisting of metasurfaces and multilayer films with high efficiency and spectral resolution. The device achieves cooperative polarization and wavelength modulation by combining the polarization modulation of the upper metasurface and the amplitude modulation of the lower multilayer film and the angular coupling between them. The polarization modulation is achieved by establishing the amplitude modulation of polarization by using the Jones matrix of a dual-atomic interference model. Polarization selection and wavelength separation at different emission angles are achieved by modulating the transmission phase. The narrow-band transmission of specific polarization states is enabled by the angular coupling between the metasurface and the multilayer FP cavity. This device can control the spectral resolution within 10 nm in the operating bandwidth with an average efficiency of 75%. It can be operated at other wavelength bands. We discussed the fabrication process for the on-chip polarized spectral imager to facilitate practical applications. As a potential scheme for polarization spectral detection, our design combines high efficiency and spectral resolution and may have additional applications in diagnostics, remote sensing, and other fields.

## Figures and Tables

**Figure 1 nanomaterials-13-02624-f001:**
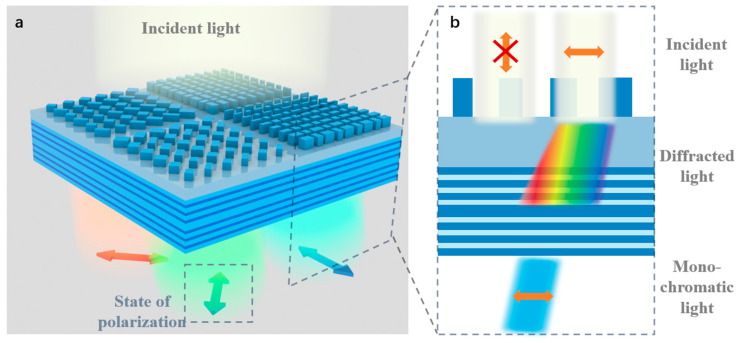
(**a**) Schematic of the proposed polarization spectral detector with multiple mosaic channels. (**b**) Schematic diagram of light propagation.

**Figure 2 nanomaterials-13-02624-f002:**
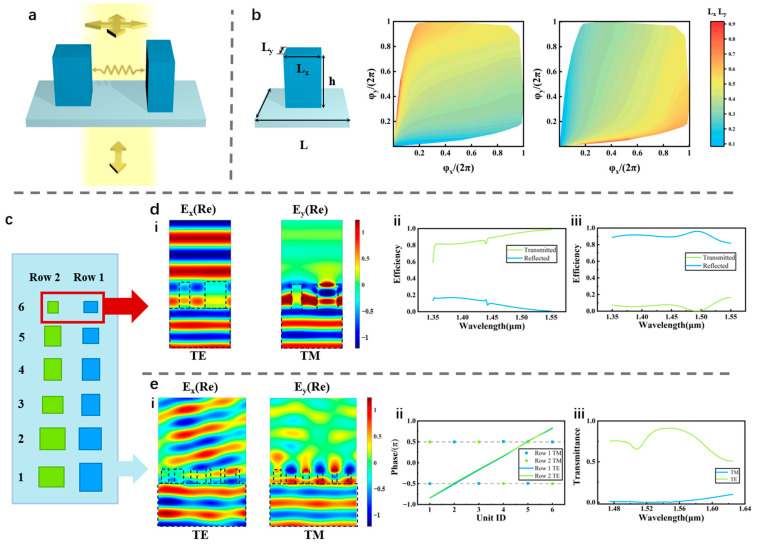
Design of the top-layer metasurface. (**a**) Schematic of polarization selection via Jones matrix modulation based on the interference of multiple meta-atoms. (**b**) Schematic diagram of a rectangular nanopillar with structural parameters and a reorganized database based on the simulation results of the phase of transmission φ_x_ and φ_y_ for different values of L_x_ and L_y_ of a single nanopillar. (**c**) Schematic of polarization selective diffractive metasurface (top view). (**d**) (**i**) Distribution of electromagnetic fields of a polarization unit under TE and TM incident light and their transmission and reflection curves in a broad wavelength range (**ii**,**iii**). (**e**) (**i**) Distribution of electromagnetic fields of the metasurface under TE and TM incident light. (**ii**) The phase profile of the metasurface under TE and TM incident light. (**iii**) Transmittance efficiency curve of TE (1st order) and TM incident light.

**Figure 3 nanomaterials-13-02624-f003:**
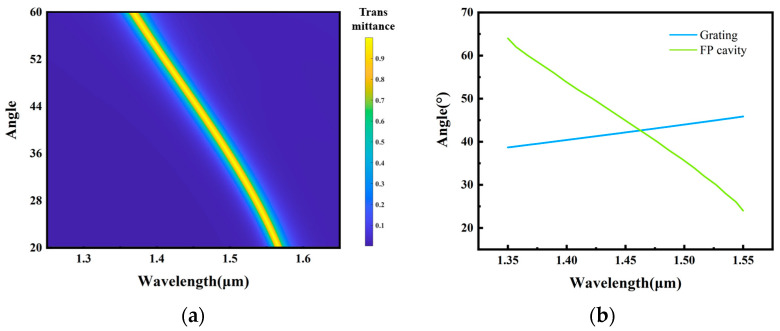
(**a**) FP cavity transmission peak at different incident angles (20° (1.57 μm) to 60° (1.35 μm)). (**b**) Relationship between the wavelength and the angle for the FP cavity and the metasurface. The intersection indicates that only a single wavelength can pass through the FP cavity.

**Figure 4 nanomaterials-13-02624-f004:**
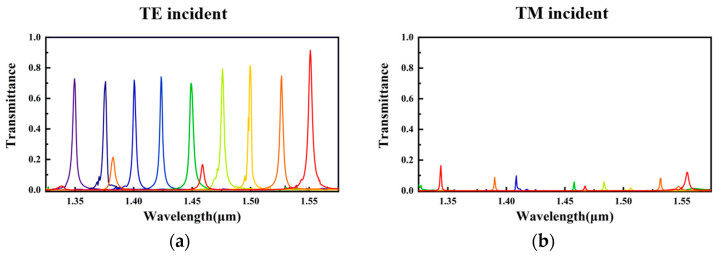
Simulation result of the on-chip polarized spectral detector. (**a**) The transmission spectra of the nine-channel wavelength bands under TE-polarized incident light. (**b**) The transmission spectra under TM-polarized incident light. The transmission of TM-polarized light is well suppressed.

**Figure 5 nanomaterials-13-02624-f005:**
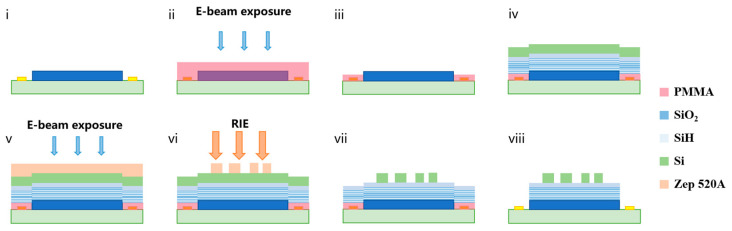
Fabrication process of the proposed device. (**i**) Schematic of the detector with the pixels and electrodes. (**ii**) Spin-coating of PMMA photoresist. (**iii**) Overlay e-beam lithography and development. (**iv**) Deposition of multiple thin films by PVD. (**v**) Spin-coating ZEP-520A photoresist, overlay e-beam lithography, and development. (**vi**) Etching of the top Si layer and remaining photoresist by RIE. (**vii**) Removing the remaining photoresist on the top Si layer (**viii**) Removing the remaining photoresist from the electrodes.

## Data Availability

Data underlying the results presented in this paper are not publicly available at this time but may be made available by the authors upon reasonable request.

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
