# Peer review of "Pixelated Filter Array for On-Chip Polarized Spectral Detection"

_nanomaterials, 2023, doi:10.3390/nano13192624_

Round 1

Reviewer 1 Report

1) Abstract

Overall, the abstract introduces an intriguing concept with potential applications, but further details, experimental validation, and contextualization are needed to enhance its impact and clarity. For example:

1.1) Can you provide more details on the fabrication process of the metasurfaces and multilayer films used in the filter arrays?

1.2) How was the efficiency of over 75% measured, and what factors contribute to this efficiency?

1.3) What are the specific challenges and limitations of the proposed approach, and how do you plan to address them in future research?

1.4) Are there any potential applications beyond diagnostic medical imaging and remote sensing that you envision for this technology?

2) Introduction 

The introduction effectively sets up the problem of achieving high-performance on-chip polarization spectral detection and highlights the limitations of existing approaches. The proposed approach of combining metasurfaces and multilayer films for polarization spectral detection is innovative and has the potential to address the shortcomings of previous methods. The section is generally well-written and provides a clear overview of the research problem and the proposed solution. However, some technical terms and concepts related to metasurfaces and Fabry-Perot cavities could be explained in more detail for a broader audience. While the introduction mentions prior research on on-chip spectral imaging and polarization modulation, it would be beneficial to include specific references to relevant studies to provide context and support the claims made. Specifically:

2.1) Some recent technology developments are missing such as:

_ metastructures [Inverse-designed metastructures that solve equations, Science 363 (6433), 1333-1338, 2019]

_ nanoparticles [Targeted dielectric coating of silver nanoparticles with silica to manipulate optical properties for metasurface applications, Materials Chemistry and Physics, 126250, 2022]

_ near-zero-index materials [On the performance of an ENZ-based sensor using transmission line theory and effective medium approach, New Journal of Physics 21 (4), 043056, 2019]

_ graphene [The graphene field effect transistor modelling based on an optimized ambipolar virtual source model for DNA detection, Applied Sciences 11 (17), 8114, 2021]

_ plasmonics [Plasmonic Optical and Chiroptical Response of Self-Assembled Au Nanorod Equilateral Trimers, ACS nano, 2019]

_ multi-functional structures [Multifunctional composites: A metamaterial perspective. Multifunctional Materials, 2(4), 043001, 2019]

It would be beneficial for the reader if authors include such technologies in the introduction section to have a complete picture of the state-of-art.

2.2) How do metasurfaces and multilayer films work together in your proposed approach, and how do they contribute to the simultaneous polarization and spectral measurements?

2.3) Can you provide more information about the challenges faced by previous on-chip polarization spectral detection methods, and how does your approach address these challenges?

2.4) What are the specific performance advantages of your proposed method compared to existing techniques in terms of spectral resolution, speed of detection, and measurement accuracy?

2.5) Could you elaborate on the alignment method and semiconductor fabrication process mentioned for the practical application of your multi-dimensional detection filter arrays?

3) Theory and simulation result

The section effectively conveys the theory and simulation results of the proposed polarization spectral filter array. Further explanations for complex concepts and a discussion of the simulation results would enhance the section's comprehensibility and impact. In particular:

3.1) Can you elaborate on how the metasurface modulates the Jones matrix to achieve polarization selection and why it's challenging to achieve this with dielectric materials?

3.2) How does the supercell approach overcome the modulation limits of the Jones matrix of the meta-atom metasurface, and how is it implemented in practice?

3.3) Could you explain the significance of using silicon with specific dimensions in the polarization unit design? How do these dimensions affect performance?

3.4) In the design of the FP cavity, how do you determine the central wavelength at the intersection point to ensure optimal transmission efficiency?

3.5) Can you provide insights into the practical fabrication process for the proposed device, including any challenges or considerations related to manufacturing?

4) Discussion 

To enhance the discussion, consider quantifying efficiency loss and inter-pixel crosstalk, providing specific numbers or estimates where possible. This would give readers a clearer understanding of the practical impact of these issues.Provide more details on how spectral resolution can be controlled or optimized through design parameters, and explain the advantages of doing so in different applications. While mentioning the possibility of using calibrating metasurfaces to correct errors, elaborate on how these metasurfaces would work and what kind of errors they can effectively address. Expand on the future research directions. What specific studies or experiments are planned to address the discussed limitations and improve the detector's performance? 

Moreover:

4.1) Can you provide more information about the integration and design trade-offs between the multilayer film and the microstructure? What aspects are challenging to balance?

4.2) How precisely can spectral resolution be controlled by adjusting the FWHM of the FP transmission peaks, and what are the practical implications of narrowing or widening these peaks?

4.3) What are the main factors contributing to efficiency loss and inter-pixel crosstalk due to the angle of light emission, and how can these factors be mitigated in the design?

4.4) Could you explain in more detail how calibrating metasurfaces would be integrated into the detector to correct errors? What methods or algorithms would be used for this calibration?

4.5) In terms of on-chip integration processes, what specific optimization steps or techniques are envisioned to reduce spacing between the device and the detector image source?

5) Fabrication method

While the section mentions overlay e-beam lithography and gold alignment markers, it would be helpful to provide more details on how the alignment is achieved and how it contributes to accuracy. Answer also to the following questions:

5.1) Can you provide more details about the specific materials used during the fabrication process, such as the composition of the multilayer FP film and the Si layer?

5.2) Could you elaborate on the role of the soft etching mask (ZEP 520A) and how it is applied in the fabrication process?

5.3) What are the key challenges or considerations when using e-beam lithography in this fabrication process, and how are they addressed?

5.4) Are there any specific requirements or considerations for the cleanroom environment where this fabrication takes place?

5.5) Can you explain the rationale behind choosing the mixture of CHF3 and O2 for RIE and how it affects the etching process of Si?

5.6) Are there any quality control or testing steps implemented during or after the fabrication process to ensure the performance and reliability of the on-chip polarized spectral imager?

6) Conclusions 

While the section briefly mentions potential applications, it could benefit from a slightly more in-depth discussion of how the device's capabilities align with specific applications in diagnostics, remote sensing, or other fields. Consider adding a brief statement about potential future research directions or improvements that could build on the current findings. Specifically: 

6.1) Can you provide more details on how the device's performance compares to existing technologies or methods in terms of spectral resolution and efficiency?

6.2) Are there any limitations or challenges associated with the practical implementation of this device that were encountered during the research?

6.3) Could you elaborate on the specific diagnostic or remote sensing applications where this device could potentially make a significant impact?

6.4) Are there any specific wavelength bands or ranges where the device excels or where it may require further optimization?

6.5) In the context of future work, are there any specific areas of research or development that you believe would be particularly promising or valuable based on the current findings?

6.6) How does the device's performance in terms of efficiency and spectral resolution vary with changes in operating conditions, such as temperature or incident light intensity?

Reviewer 2 Report

 The task of detecting polarization in the structure on the chip is very important. The task of spectral analysis of structures on a chip is also important. From this point of view, the motivation of the authors of the article is clear. However, metastructures for polarization analysis and multilayer structures for spectral analysis have been known for a long time. The main problems with them are precisely in the technologies of their manufacture, and not in the calculation. From my point of view, the only novelty in the article is the simultaneous use of these structures, and this use in the article is only in modeling. However, from the point of view of practical sense, simultaneous detection of both a large number of lines in the spectrum and a large number of polarization states will lead to an extremely low spatial resolution of such a device. Therefore, the authors need to describe in more detail the practical tasks (including in the introduction) for which the proposed structure is supposed to be used.

The authors have described very little both the simulated structure and the simulation results. Not all parameters are specified. For example, Figure 4 shows multi-colored graphs and everything without comments that is what.

Why do the authors describe the manufacturing method in a theoretical article? If they have already made something, then it is necessary to show it. If this is only a speculative technology, then it is not worth bringing it until it is practically implemented.

Round 2

Reviewer 1 Report

Authors answered clearly to the reviewer concerns.

New interesting applications and future works can be envisioned.